# Individuality and convergence of the infant gut microbiota during the first year of life

Eric J. de Muinck[1] & Pål Trosvik[1]

The human gut microbiota plays a vital role in health and disease, and microbial colonization is a key process in infant development. Here, we analyze 2684 fecal specimens from 12 infants during their first year of life, providing detailed insights into the human gut colonization process. Maturation of the gut microbial community shows strong temporal structure and specific developmental stages. At 2–4 months of age, there is a period of accelerated convergence concurrent with a bloom of *Bifidobacterium*, a genus associated with metabolism of oligosaccharides found in breast milk. The end of this period coincides with the introduction of solid food, a reduction in the relative abundance of *Bifidobacterium*, and an increase in several groups of Firmicutes. Our findings highlight the dynamic nature and individuality of the gut colonization process, and the need for high-frequency sampling over an extended period when designing and interpreting infant microbiome studies.

[1] Centre for Ecological and Evolutionary Synthesis (CEES), Department of Biosciences, University of Oslo, Oslo, Norway. Correspondence and requests for materials should be addressed to P.T. (email: pal.trosvik@ibv.uio.no)

Humans are born with an essentially sterile gastrointestinal (GI) tract with microbial colonization starting during birth[1]. By the age of ~3 years, the GI microbial community has developed to form an adult-like structure[2,3]. Many claims have been made associating the infant GI microbiota with health outcomes[1,4,5], including later development of conditions such as obesity[6,7], inflammatory bowel disease[8], and type 1 diabetes[9,10]. Epidemiological studies have also linked the developing infant GI microbiota to these conditions[11–13], as well as autism[14] and allergy[15]. Although powerful, studies characterizing the development of the infant gut microbiome are usually based on fecal samples taken at relatively infrequent intervals[9,16], for a brief duration[17,18], or focused on a single infant[19,20]. There is yet no study of a cohort of healthy, born to term infants with high-frequency sampling for a sustained period. Thus, we do not currently have a clear view of the patterns that constitute normal development of the early gut microbial community. Lacking this information, clinical studies comparing health outcomes may confound "at-risk" states with natural variation and transient dynamics. Here, we provide a high-resolution view of the microbial colonization process of 12 infants, using fecal samples obtained on a near daily basis during the first year of life, including one pair of dizygotic twins and one pair of siblings born 16 months apart. Although developmental trajectories are highly individual, they all show pronounced temporal structure and non-linear dynamics. Furthermore, we observe a period of accelerated convergence, between ~60 and 130 days after birth, when the microbiotas of the infants become much more similar to one another. This period coincides with a bloom of *Bifidobacterium* spp. and a decline in several groups within the phylum Firmicutes, and concludes roughly at the time of introduction of solid food. The twins' GI microbiotas track each other closely throughout, despite of one receiving intensive antibiotics treatment towards the end of the first month of life. Our results highlight the importance of considering the individual and dynamic nature of the infant gut colonization process when designing and interpreting studies.

## Results

**Data overview.** In total, 2684 stool samples were analyzed with an average of 224 samples (range 116–267) per infant (Table 1). 16S rRNA gene amplicon sequencing resulted in 440,752,576 reads after quality trimming, paired read merging and sequence clustering, with a mean per sample read number of 164,215 ($\pm$ 54,438 s.d.). In the entire data set, we observed 1736 operational taxonomic units (OTUs) on the 97% sequence identity level (Supplementary Data 1). The number of OTUs shared between individuals followed a U-shaped distribution (Supplementary Fig. 1), with 229 OTUs observed in at least one sample in all infants, while 331 were unique to a single infant. Overall, diversity increased throughout the first year (Supplementary Figs. 2 and 3). However, these increases were neither linear nor monotonous, and most children went through intermittent periods of declining diversity.

**OTU level temporal structure.** All infants exhibited both gradual and punctuated shifts in microbiota structure (Supplementary Figs. 4–15). Often, but not always, changes were associated with periods of travel outside of Norway/Sweden (Supplementary Figs. 4, 6 and 11). The colonizing process was similar across all the infants in that it showed strong temporal structure (Fig.1). Age was the main structuring factor, as determined by regressing the first non-metric multidimensional scaling (nMDS) component of the models describing each individual infant on the number of days since birth (mean $R^2 = 0.7$ using both Bray–Curtis (BC) (Fig. 1) and weighted UniFrac (wUF) (Supplementary Fig. 16) as distance metrics; $p < 0.001$ for all tests, linear regression). Even though the microbiotas were highly dynamic during development, the infants clustered significantly by individual ($R^2 = 0.29$ and 0.27; PERMANOVA using BC and wUF distances, respectively; $p < 0.001$ for both tests) (Supplementary Fig. 17a and 18a). In addition, there was a significant common time signal as determined by regressing the first nMDS component of the model describing all the infants on the concatenation of the individual time vectors ($R^2 = 0.17$ and 0.26; linear regression using BC or wUF distances, respectively; $p < 0.001$ for both tests) (Supplementary Fig. 17b and 18b).

**Period of accelerated convergence.** To investigate whether the children developed more similar GI microbiotas as they matured, we interpolated the data series of 11 of the infants (ID8 was censored from this analysis due to sampling beginning on day 54) to equal length and computed pairwise contemporaneous BC and wUF distances (see Methods). This analysis demonstrated that the mean pairwise distances did indeed decrease over time ($R^2 = 0.2$ for both BC and wUF distances, $p < 0.001$, linear regression). However, this trend was highly non-linear, and included a pronounced period of accelerated convergence of the GI microbiotas approximately from day 60 to 130, followed by a period of increasing divergence roughly until day 200 (Fig. 2a,

### Table 1 Overview of cohort

| Infant | Gender | Mode of delivery | Formula or breast milk | Day of first sample | Day of last sample | Samples | Observed OTUs | Unique OTUs | Day of introduction of solid food |
|--------|--------|------------------|------------------------|---------------------|--------------------|---------|---------------|-------------|-----------------------------------|
| ID1 | Girl | Vaginal | BM | 2 | 355 | 241 | 808 | 17 | 186 |
| ID2 | Boy | Vaginal | F | 6 | 366 | 223 | 754 | 61 | 159 |
| ID3 | Girl | Vaginal | BM | 5 | 365 | 257 | 1029 | 35 | 165 |
| ID4[a] | Girl | Vaginal | BM | 4 | 357 | 266 | 934 | 22 | 126 |
| ID5[a] | Boy | Vaginal | BM | 2 | 367 | 253 | 999 | 43 | 116 |
| ID6 | Boy | Vaginal | Mix | 2 | 363 | 196 | 538 | 6 | 121 |
| ID7 | Boy | Vaginal | BM | 9 | 365 | 116 | 643 | 25 | 122 |
| ID8 | Girl | Vaginal | BM | 54 | 362 | 209 | 497 | 6 | 247 |
| ID9 | Girl | C-section | BM | 4 | 364 | 250 | 900 | 71 | 182 |
| ID10[b] | Boy | C-section | Mix | 6 | 365 | 203 | 867 | 21 | 172 |
| ID11[b] | Girl | C-section | Mix | 6 | 365 | 234 | 768 | 11 | 173 |
| ID12 | Girl | Vaginal | Mix | 1 | 366 | 236 | 849 | 13 | 122 |

BM = breast milk, F = formula fed, Mix = both breast milk and formula. The numer of OTUs observed are the total number of OTUs observed in an infant over the entire sampling period. The number of unique OTUs are the number of OTUs found only in that infant
[a]ID4 and ID5 are siblings born ~16 months apart
[b]ID10 and ID11 are fraternal twins

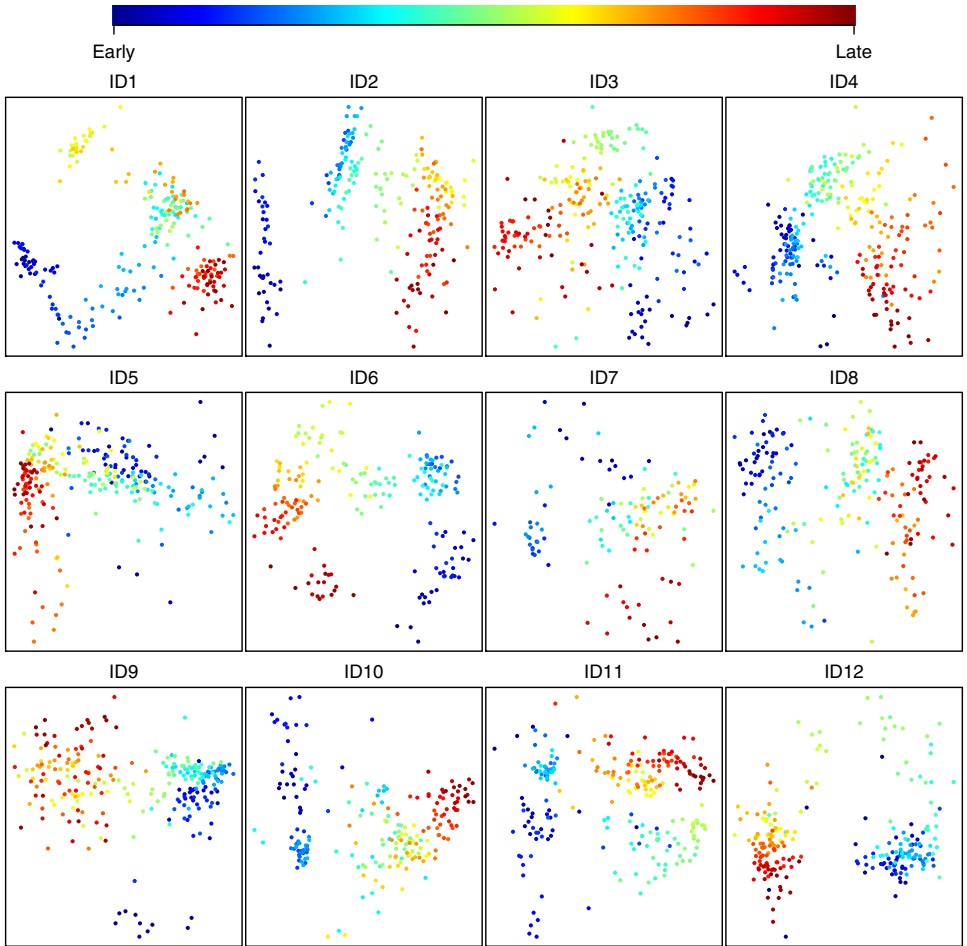

**Fig. 1** Development of the infant GI microbiota is highly structured in time. Each panel shows a non-metric multidimensional scaling (nMDS) plot based on Bray–Curtis distances, for each of the 12 infants. For each model there is a highly significant relationship between the primary axis of variation and time since birth (for all tests $p \ll 0.001$, mean $R^2 = 0.7$ (range 0.47–0.86), linear regression). Dots are colored according to the bar above the panels. Early and late refer to the order in which the samples were collected (see Table 1 for days of first and last sampling)

Supplementary Figs. 19–21). Convergence was primarily driven by an increase in the mean relative abundances of Actinobacteria and a concurrent decrease in Firmicutes (Fig. 2b, Supplementary Fig. 22). Correlations between the relative abundances of the twenty most abundant OTUs (78.3% of total abundance) and the pairwise contemporaneous BC distances from day 60 to 130 were computed in order to identify the main genera associated with convergence (Supplementary Table 1). The bloom of Actinobacteria was dominated by *Bifidobacterium longum/breve* (OTU1, 94.4% of total Actinobacterial abundance, Supplementary Fig 23a). In contrast, several OTUs were affected by the collapse of the Firmicutes (Supplementary Fig 23b). The conclusion of the accelerated convergence period, and the subsequent divergence period, coincided roughly with introduction of solid food (Table 1).

**Phylum level dynamics**. As expected, the GI microbiota of each of the infants was dominated by four main phyla: Bacteroidetes (mean 32.3% range 1.1–58.7%), Actinobacteria (mean 16.1% range 0.01–33.7%), Firmicutes (mean 35.4% range 19.2–72.9%), and Proteobacteria (mean 15.5% range 6.6–23.2%). However, within each child, relative abundances of these phyla followed complex temporal trajectories that differed markedly between individuals (Fig. 3, Supplementary Figs. 24–27). In accordance with previous studies[16,21], we observed a general decline in Actinobacteria and Proteobacteria, and increasing dominance of Bacteroides and

Firmicutes towards the first birthday. A high prevalence of Actinobacteria, is considered a hallmark of the early infant microbiome[2], with formula fed infants thought to have reduced colonization by *Bifidobacterium* spp.[22]. Interestingly, even though ID1 was breastfed during the first 6 months, she had very low relative abundances of Actinobacteria (Fig. 3a, Supplementary Fig. 24).

Bacteroidetes are strongly associated with the GI microbiomes of human infants and adults, as well as other mammalian species[2,23]. In contrast to the other infants in this study, ID6 had extremely low relative abundances of Bacteroidetes (Fig. 3a, Supplementary Fig. 25), without a single OTU among the 40 most abundant classified to that phylum. In order to find which infants had the most similar phylum level dynamics, we carried out hierarchical clustering based on mean pairwise dynamic time warp distances (see Methods). This analysis identified two main branches (Fig. 3b) with ID6 separated on a deep branch. The dizygotic twins (ID10 and ID11), who are no more genetically similar than a sibling pair, clustered tightly, suggesting that they tracked each other well. However, the two siblings born 16 months apart (ID4 and ID5) clustered on separate main branches.

**Comparison of the twins**. One of the dizygotic twins (ID10) was hospitalized due to *Streptococcus* serogroup B β-hemolytic (*S. agalactiae*) bacteremia and was treated with intravenous (iv)

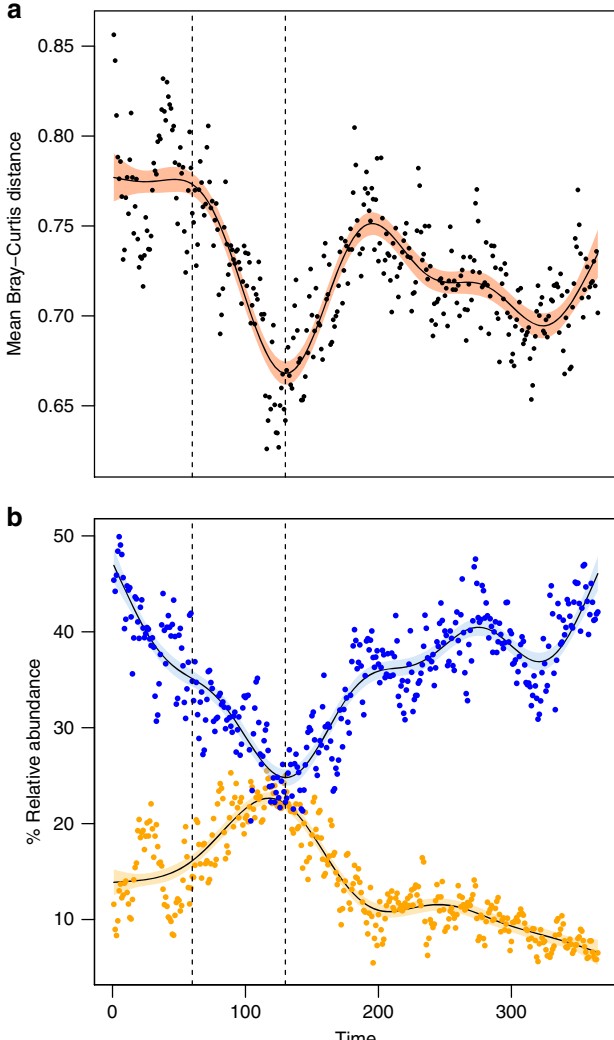

**Fig. 2** The infant GI microbiota goes through a period of accelerated convergence approximately between days 60 and 130 after birth. All data series were interpolated to equal length (365 pseudo-days) for this analysis. Dots show the observed values while lines show fitted generalized additive models ($p \ll 0.001$ for all model fits). **a** Mean pairwise contemporaneous Bray–Curtis distances between 11 infants* ($R^2 = 0.65$). **b** Relative abundances of *Bifidobacterium longum/breve* (OTU1) (yellow, $R^2 = 0.79$) and Firmicutes (blue, $R^2 = 0.74$). Shaded bands represent 95% confidence limits for the fitted models. Dotted vertical lines indicate the window of convergence (days 60–130). * ID8 was excluded from this analysis since sampling did not commence until day 54

cloxacillin and gentamycin on day 22, then switched to iv ampicillin and gentamycin from day 23 to day 28 (Fig. 4). He was treated briefly with iv. cefotaxime on day 29–30 and then oral penicillin until day 33. In spite of the treatment, the twins' colonization patterns tracked each other closely (Fig. 4, Supplementary Fig. 28) and were much more similar to each other than either were to any of the other infants (t-test, $p < 0.001$, Supplementary Fig. 29). Both infants were colonized by several OTUs classified as *Streptococcus* and ID10 experienced a bloom of *S. agalactiae* directly before the onset of treatment, while a corresponding bloom was not observed in the twin sister (Supplementary Fig. 30). However, antibiotic treatment did not have any apparent effects on subsequent streptococcal colonization and blooms. In fact, the dynamics the four most abundant OTUs

classified as *Streptococcus* were remarkably similar in the twins throughout the study period (Supplementary Fig. 30).

## Discussion

In this study, we analyzed near daily fecal samples collected from a cohort of infants during the first year of life, providing a high-resolution standard reference of the infant gut colonization process. Our findings were largely in accordance with previous reports, e.g., phylum level colonization patterns[16,21] and gradual increase in diversity[9,19]. However, the non-linear properties of these processes have, to our knowledge, not been described previously and would not have been observable without high-frequency sampling. Convergence of the infant gut microbiota towards decreasing beta-diversity, over time scales from months to years, has also been reported[2,24]. The defined period of accelerated convergence reported here demonstrates a pattern, general across the cohort, of fine-scale temporal dynamics, and the potential importance of dense longitudinal sampling in order to describe important phenomena in the infant gut colonization process.

Although antibiotic treatment has been shown to disrupt the infant gut microbiome, both immediate and long term effects are not well understood[3]. For example, one study of the gut microbiota in preterm infants found that administration of antibiotics in the neonatal intensive care unit did not significantly affect the long term development of the GI microbiota[25]. It is possible, although this would need to be substantiated in a larger study with proper controls, that the apparent lack of effect on the GI microbiota in the case of the twins in our study could be the result of near constant re-seeding through contact with an age-matched sibling.

Our results emphasize the importance of longitudinal sampling, of an appropriate resolution, in order to properly describe and compare microbiotas of human infants. Similar studies, of wider scope in terms of sampling duration, cohort size, locations and lifestyles, should be undertaken in order to gain an even more profound understanding of the human developmental process. This would allow for rational design of studies linking individual developmental trajectories to relevant health outcomes.

## Methods

**Sample collection**. All recruited infants were born to term in Oslo, Norway. All parents signed a participation consent form approved by the Norwegian Regional Committee for Medical and Health Ethics, project 2014/656, and the study was in compliance with the Helsinki Declaration. Most fecal samples were frozen at −20 °C immediately upon collection, pending transport, on dry ice, to a −80 °C storage facility at the University of Oslo. A subset of fecal samples was taken while families were traveling. In these cases the samples were stored in 95% alcohol. These samples are the following: ID3 days 127–132, 153–177, 198–273, and 311–365; ID5 days 87–167; ID10 days 80–169 and 229–366; ID11 days 80–169 and 229–366. ID10 was hospitalized due to *Streptococcus* serogroup B β-hemolytic (*S. agalactiae*) bacteremia and was treated with intravenous (iv) cloxacillin (200 mg per day) and gentamycin (24 mg per day) on day 22, then switched to iv ampicillin (200 mg per day) and gentamycin (24 mg per day) from day 23 to day 28. He was treated briefly with iv. cefotaxime (500 mg per day) on day 29–30 and then oral penicillin until day 33 (280 mg per day). ID12 was treated with the antifungal drug Mycostatin (400,000 IU per day) taken orally from day 37–49. None of the other infants were given antimicrobial drugs during the sampling period. ID10 and 11 had a cat and a dog as house pets. ID9 had a pet rabbit.

**DNA extraction and Illumina sequencing**. DNA extraction from all samples was carried out with the PowerSoil 96 well DNA isolation kit (MO BIO Laboratories Inc., Carlsbad, CA, USA), per instructions provided by the manufacturer. Library preparation for Illumina sequencing of the V4 region of the 16S rRNA gene was carried out according to de Muinck et al.[26], using the primers 515F (GTGYCAG CMGCCGCGGTAA) and 806R (GGACTACNVGGGTWTCTAAT). Sequencing was done at the Norwegian Sequencing Centre (http://www.sequencing.uio.no/) on an Illumina HiSeq 2500 apparatus (Illumina, San Diego, CA, USA) using the 2 × 250PE rapid run mode and 10% PhiX spike-in. Sequencing resulted in a total of 440,752,576 reads after quality filtering, paired read merging and sequence clustering. The mean per sample read number was 164,215 ( ± 54,438 s.d.).

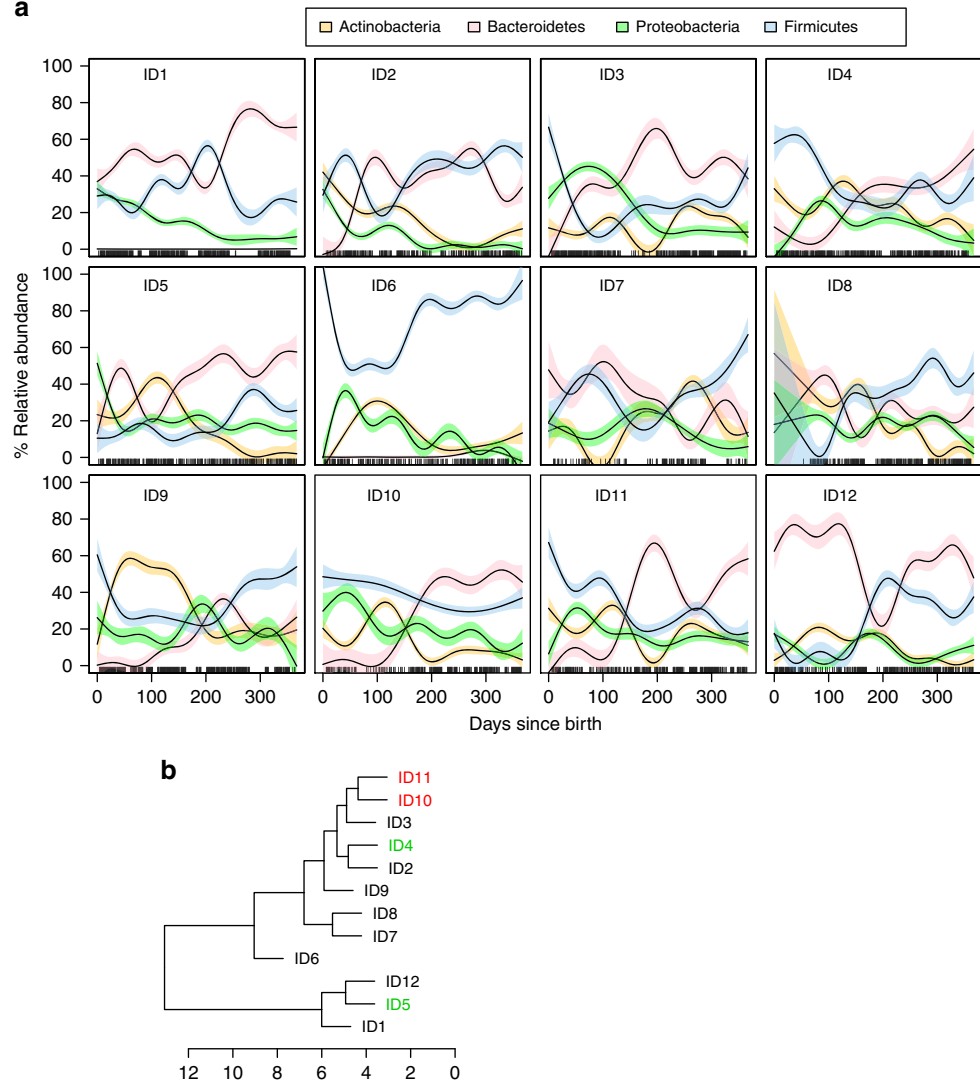

**Fig. 3** Phylum level dynamics of the GI microbiota over the first year of life in the 12 infants cluster on two main branches. **a** Relative abundances of the four main phyla in each infant. Lines are fitted generalized additive models and colored bands are 95% confidence limits for smooth terms of the fitted models (see color code). Rugs along the x-axis indicate days for which we had samples. **b** Hierarchical clustering based on mean dynamic time warp distances between the time series in **a**. The two twins are indicated in red, while the siblings born 16 months apart are indicated in green. The axis beneath the tree indicates number of distance units separating the infants

**Sequence data processing**. Low quality reads were trimmed and Illumina adapters were removed using Trimmomatic v0.36[27] with default settings (i.e., the 3′ end of a read was trimmed if the average quality per base dropped below 15 using a 4-base sliding window). Reads mapping to the PhiX genome (NCBI id: NC_001422.1) were removed using BBMap v36.02[28]. De-multiplexing of data based on the dual index sequences was carried out using custom scripts[26]. Internal barcodes and spacers were removed using cutadapt v1.4.1[29] and paired reads were merged using FLASH v1.2.11[30] with default settings. Further processing of sequence data was carried out using a combination of vsearch v2.0.3[31] and usearch v9.2.64[32]. Specifically, dereplication was performed with the "derep_fulllength" function in vsearch with the minimum unique group size set to 5, in order to eliminate artefactual OTUs. Operational taxonomic unit (OTU) clustering, chimera removal, taxonomic assignment and OTU table building were carried out using the uparse pipeline[33] in usearch. Taxonomic assignment to the genus level was done against the RDP-15 training set. Classification to the species level was done by BLASTing[34] against the GenBank 16 S rRNA gene database. OTUs based on consensus sequences shorter than 200 bp and longer than 260 bp were removed from the data. OTUs with a domain-level assignment probability <0.95 were also dropped.

**Statistical analysis**. Between sample differences in sequencing library size were normalized by common scaling[35]. This entails multiplying all OTU counts for a given library with the ratio of the smallest library size in the entire data set to the

size of the individual library. This procedure replaces rarefying (i.e., random sub-sampling to the lowest number of reads) as it produces the library scaling one would achieve by averaging over an infinite number of repeated sub-samplings. Library size scaling was carried out using a smallest library size of 50,593 reads. Read counts were then rounded to the closest integer.

All statistical tests were done in R[36]. Permutational Multivariate Analysis of Variance Using Distance Matrices (PERMANOVA) was carried out using the "adonis" function in the "vegan" package[37], using Bray–Curtis or weighted UniFrac distances, with 1000 permutations. Weighted UniFrac distances were computed using the "GUniFrac" function in the "GUniFrac" package[38], with the α parameter controlling the weight on abundant lineages set to 0.5, as recommended in the package documentation. For constructing the phylogeny upon which the Unifrac distances were based all OTU sequences were aligned using MUSCLE[39] and a neighbor-joining tree[40] was constructed using MEGA v7.0.26[41]. Non-metric multidimensional scaling (NMDS) of Bray–Curtis and weighted UniFrac distance matrices were carried out using the "isoMDS" function in the "MASS" package[42]. Generalized additive models (GAMs) were computed using the "gam" function in the "mgcv" package[43], with 9 degrees of freedom for estimating the smooth terms in order to allow for significant non-linearity in the relationship between response and predictor. Dynamic time warp (DTW) distances between time series were computed using the "dtw" function in the "dtw" package[44], using Euclidian distances for the pointwise local distance function, the "symmetric2" step pattern and no global constraint on the warping path. Specifically, in order to obtain a

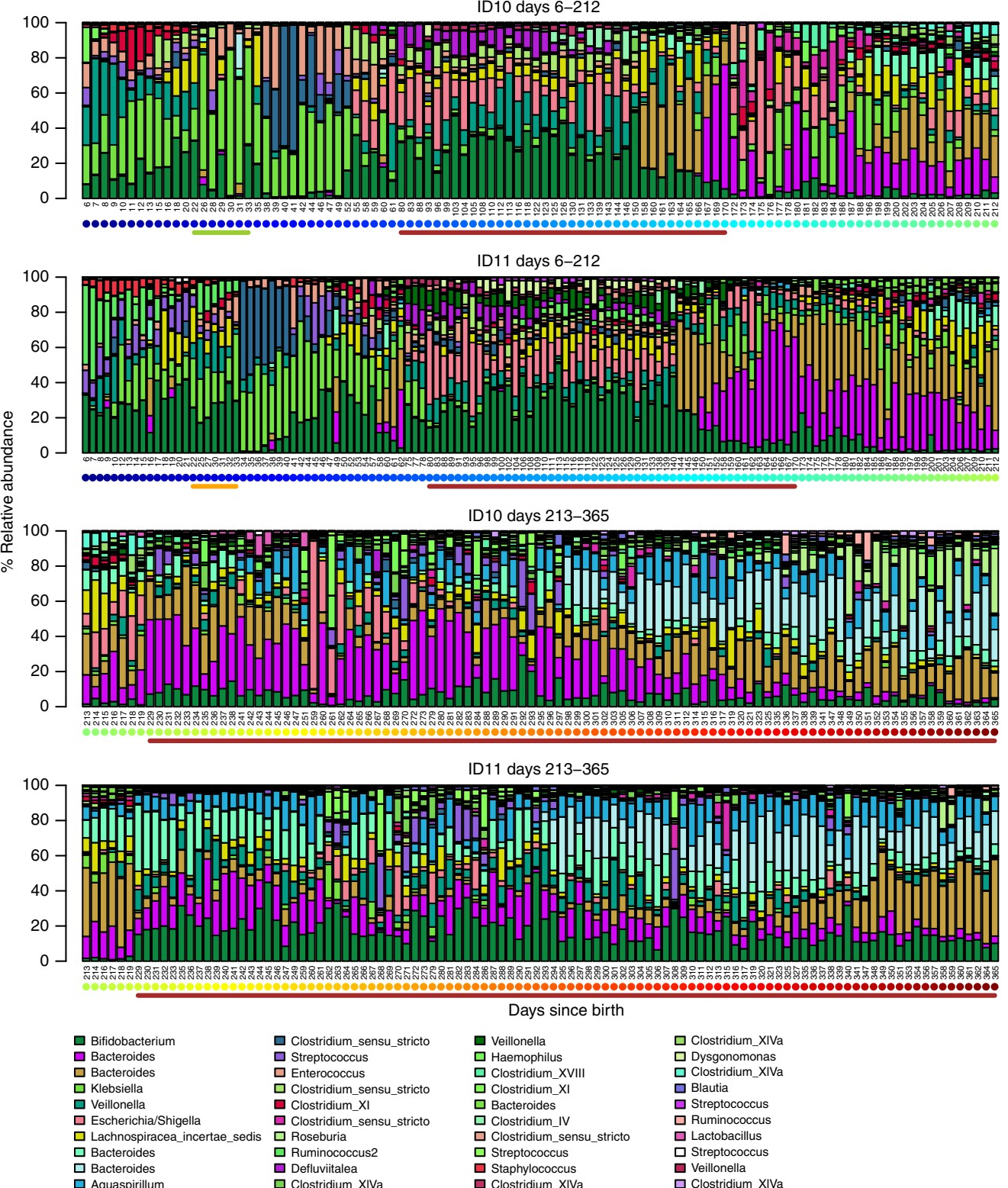

**Fig. 4** Two fraternal twins track each other closely over the first year of life, in spite of one receiving intensive antibiotic treatment. Each panel shows relative abundances of the 40 most abundant genera in ID10 and ID11, with each bar representing a sample and each colored bar segment representing the relative abundance of the genus indicated in the colour key below. The green line beneath the top panel indicates the period during which ID10 received antibiotic treatment. The corresponding period for ID11 is indicated by the orange line beneath the second panel. The brown horizontal lines indicate time spent outside Norway. Dots beneath the days since birth (x-axes) indicate the time of sampling (dark blue to dark red = early to late)

single distance measure between each pair of infants we used the mean of the normalized (to series length) pairwise DTW distances based on the time series of the four main phyla (Actinobacteria, Bacteroidetes, Firmicutes and Proteobacteria). This was done to make sure we were comparing time series of taxonomic groups found in all individuals. Clustering of the DTW distance matrix was done using the "hclust" function in the "stats" package[42], with the complete linkage method. Interpolation of all individual data series to equal length (365 data points) was done using the function "spline" in the "stats" package, with the 'fmm' method for fitting an exact cubic polynomial through the four points at each end of a time series[45]. Negative interpolated values were set to zero. Interpolation was carried out to facilitate point-by-point comparisons between time series of unequal length. ID8 was excluded from this analysis since for this infant sampling did not commence until day 54 after birth. Specifically, the interpolated time series were used to compute Bray–Curtis and unweighted UniFrac distances for each of 365 "pseudo-days", i.e., interpolated data points, between all possible pairs of infants. For each infant the mean distance to all other infants was computed for all "pseudo-days", producing a 365 data point vector of average distances for each of the eleven infants. Finally, the grand mean of the eleven vectors was computed, resulting in one 365 data point vector describing the average dissimilarity between the infants. The interpolated data series were also used to compare the mean distances between the two twins, and between each twin and the other infants.

**Data availability**. All sequence data used in this study are available in the NBCI Sequence Read Archive under SRA accession codes SRP141136 (ID1), SRP141176 (ID2), SRP141301 (ID3), SRP141326 (ID4), SRP141372 (ID5), SRP142048 (ID6), SRP142071 (ID7), SRP142093 (ID8), SRP142140 (ID9), SRP142235 (ID10), SRP142291 (ID11), and SRP142429 (ID12). The IDs in parenthesis relate to the identifier of each infant in the study and are not part of the accession codes.

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

## Acknowledgements

We thank our study volunteers for their participation and dedicated work. We would also like to thank S.W. Qiao and E.K. Rueness for helpful notes on the manuscript. This study was supported by the Research Council of Norway grant 230796/F20.

## Author contributions

E.J.D. and P.T. designed the study, performed the experimental work, performed computational analyses, and prepared the manuscript.

**Additional information**

**Competing interests:** The authors declare no competing interests.

