## [Peer Review File · Nature Communications]

Reviewers' comments:

Reviewer #1 (Remarks to the Author):

This is a well presented easy to follow paper the strength is the virtually daily sampling, although there are still only 12 infants it would help to know whether infants had other 'exposures' and when weaning etc occurred the fundamental messages have been shown in preterm infants - that of immense individual variation, but eventual convergence, almost regardless of exposures/illnesses along the way - it might be helpful to reference this work (13. Stewart C, Skeath T, Nelson A, Marrs E, Perry J, Cummings C, Berrington J and Embleton N Preterm gut microbiota and metabolome following discharge from intensive care. Scientific Reports 5:17141 | DOI: 10.1038/srep17141 Nov 1, 2014

Reviewer #2 (Remarks to the Author):

This is a robust study on the early dynamics of the infant gut microbiota. By using an intensive sampling regime with almost daily fecal samples over the duration of a 3 years, this study provides the most detailed temporal insight into the dynamics of the gut microbiota. Clearly, this alone is reason to see this dataset published. However, that being said, the manuscript is very limited in novelty, and although this is probably the cleanest and most robust presentation of early microbiome dynamics, there is virtually nothing in this manuscript that I did not already know from previous, admittedly less comprehensive studies. The study has further limitations in that it has not been specifically designed to actually determine the factors that influence microbiota assembly. This leaves the study merely descriptive, and several of the really interesting findings (e.g. the high similarity of the dynamics in twins despite one of them receiving antibiotics) remain inconclusive, which is a real shame. If this study would have been designed to systematically compare twins with unrelated infant, and/or compare C-sections with vaginal births, and/or formula fed infants with breast fed, this could have been an outstanding contribution. As it is, it is a great presentation of the dynamics of the early infant microbiota, but I see nothing here that really provides an novel information.

Specific points:

Line 33: It is not clear what 'highly structured' means here, especially given that every infant shows a unique set of developmental stages.

Line 38: The genus Bifidobacterium is not really replaced by other taxa. It becomes reduced.

Line 40/41: Clearly, it could be that the presence of a twin mitigates the effect of antibiotics, but without a larger sample size and a strict comparison with the effect of similar antibiotic regimens in infants without twins, one cannot really make such a conclusion.

Line 41: So the abstract finishes by stating that this study highlights both individuality and dynamics of the gut colonization process. So both is known for a long time, including key findings of this study here that diversity increases while individuality decreases (see <https://www.ncbi.nlm.nih.gov/pubmed/25250028>). What did we learn new?

Lines 75-76: This is a really weird statement. So the process was similar in that it was distinct?

Lines 118-123: This is very descriptive information and neither novel nor helpful, as it is based on observation made in two infants. Bifidobacteria are on average higher in breast fed infants, but it is known for decades that there is substantial individuality.

Lines 125-128: Same here. This is completely descriptive.

Lines 131-134: As described above, this is really interesting observation, but it is inconclusive as it has only been observed in one single twin pair.

Lines 141-156: Same here. This is super interesting, but without a systematic comparison on the effects of antibiotics in twins versus non-twins, firm conclusions cannot be made.

Point-by-point response to reviewer comments

Reviewers' comments:

Reviewer #1 (Remarks to the Author):

This is a well presented easy to follow paper the strength is the virtually daily sampling, although there are still only 12 infants it would help to know whether infants had other 'exposures' and when weaning etc occurred the fundamental messages have been shown in preterm infants - that of immense individual variation, but eventual convergence, almost regardless of exposures/illnesses along the way - it might be helpful to reference this work (13. Stewart C, Skeath T, Nelson A, Marrs E, Perry J, Cummings C, Berrington J and Embleton N Preterm gut microbiota and metabolome following discharge from intensive care. Scientific Reports 5:17141 | DOI: 10.1038/srep17141 Nov 1, 2014

Response:

We greatly appreciate the comments made by the reviewer. In the revised we have listed the day of first introduction of solid food, i.e. the onset of the weaning period, in table one. Apart from the one twin receiving intensive antibiotics treatment, as stated in the original manuscript, the only other infant to receive any form of antimicrobial therapy was ID12, who was briefly treated with the antifungal Mycostatin. Also three of the infants grew up with exposure to house pets. All of this information is now included in the revised Methods (lines 354-360). In the revised manuscript the suggested reference is discussed and cited on lines 149-152.

Reviewer #2 (Remarks to the Author):

This is a robust study on the early dynamics of the infant gut microbiota. By using an intensive sampling regime with almost daily fecal samples over the duration of a 3 years, this study provides the most detailed temporal insight into the dynamics of the gut microbiota. Clearly, this alone is reason to see this dataset published. However, that being said, the manuscript is very limited in novelty, and although this is probably the cleanest and most robust presentation of early microbiome dynamics, there is virtually nothing in this manuscript that I did not already know from previous, admittedly less comprehensive studies. The study has further limitations in that it has not been specifically designed to actually determine the factors that influence microbiota assembly. This leaves the study merely descriptive, and several of the really interesting findings (e.g. the high similarity of the dynamics in twins despite one of them receiving antibiotics) remain inconclusive, which is a real shame. If this study would have been designed to systematically compare twins with unrelated infant, and/or compare C-sections with vaginal births, and/or formula fed infants with breast fed, this could have been an outstanding contribution. As it is, it is a great presentation of the dynamics of the early infant microbiota, but I see nothing here that really provides an novel information.

Response:

We appreciate the positive comments made by the reviewer. We feel that the level of detail at which this study was conducted provides a novel perspective of the infant developmental process. Due to the density of sampling we are able to document significant non-linear phenomena that would not be observable otherwise, such as the defined window of accelerated convergence, as well as the fine-scale dynamics of bacterial populations and OTU diversity. For designing and interpreting studies of the infant gut microbiota this kind of information can be of great value.

However, the reviewer is correct in that this study was not designed to evaluate the effects of such factors as antibiotic usage, delivery mode and diet. Rather, we set out to explore colonization

dynamics of the infant gut at a great level of temporal detail. Thus, we believe that the scope of our study in and of itself represents considerable novelty.

Specific points:

Line 33: It is not clear what 'highly structured' means here, especially given that every infant shows a unique set of developmental stages.

Response:

In the revised abstract we have changed this sentence to emphasize that we were referring to temporal structure.

Line 38: The genus Bifidobacterium is not really replaced by other taxa. It becomes reduced.

Response:

In the revised manuscript we have re-written this sentence in order to clarify the point made by the reviewer.

Line 40/41: Clearly, it could be that the presence of a twin mitigates the effect of antibiotics, but without a larger sample size and a strict comparison with the effect of similar antibiotic regimens in infants without twins, one cannot really make such a conclusion.

Response:

We agree that this claim needs to be investigated using a larger sample size and proper controls. In the revised manuscript this claim has been removed from the abstract.

Line 41: So the abstract finishes by stating that this study highlights both individuality and dynamics of the gut colonization process. So both is known for a long time, including key findings of this study here that diversity increases while individuality decreases (see <https://www.ncbi.nlm.nih.gov/pubmed/25250028>). What did we learn new?

Response:

We agree that increasing diversity and an overall decrease in individuality have been described previously. However, these phenomena have not previously been described at high temporal resolution, and thus the non-linear nature of gut colonization process has not been properly described prior to this study. In the revised manuscript we have altered the end of the abstract to reflect this.

Lines 75-76: This is a really weird statement. So the process was similar in that it was distinct?

Response:

In the revised manuscript we have altered the sentence to clarify that we meant that the colonization process was similar across infant in that there was strong temporal structure.

Lines 118-123: This is very descriptive information and neither novel nor helpful, as it is based on observation made in two infants. Bifidobacteria are on average higher in breast fed infants, but it is known for decades that there is substantial individuality.

Response:

These sentences have been removed in the revised manuscript.

Lines 125-128: Same here. This is completely descriptive.

Response:

We agree this is a descriptive statement. However, we think that this finding is notable enough that it merits mention in the main text.

Lines 131-134: As described above, this is really interesting observation, but it is inconclusive as it has only been observed in one single twin pair.

Response:

The statement referred to is about the branching pattern of DTW clustering of the 12 infants. While we only have one set of twins in the study, our statement merely points out that they cluster together (Figure 3b). We do not attempt to draw any conclusion other than that the twins have similar gut microbiotas, which to us does not feel controversial.

Lines 141-156: Same here. This is super interesting, but without a systematic comparison on the effects of antibiotics in twins versus non-twins, firm conclusions cannot be made.

Response:

We agree with this comment. We have added a statement saying that any claims need to be substantiated in larger, specially designed studies (lines 152-153).